# Kaempferol as an Alternative Cryosupplement for Bovine Spermatozoa: Cytoprotective and Membrane-Stabilizing Effects

**DOI:** 10.3390/ijms25074129

**Published:** 2024-04-08

**Authors:** Štefan Baňas, Eva Tvrdá, Filip Benko, Michal Ďuračka, Natália Čmiková, Norbert Lukáč, Miroslava Kačániová

**Affiliations:** 1Institute of Biotechnology, Faculty of Biotechnology and Food Sciences, Slovak University of Agriculture in Nitra, Tr. A. Hlinku 2, 94976 Nitra, Slovakia; stfnbanas@gmail.com (Š.B.);; 2AgroBioTech Research Centre, Slovak University of Agriculture in Nitra, Tr. A. Hlinku 2, 94976 Nitra, Slovakia; 3Institute of Horticulture, Faculty of Horticulture and Landscape Engineering, Slovak University of Agriculture, Tr. A. Hlinku 2, 94976 Nitra, Slovakia; 4Institute of Applied Biology, Faculty of Biotechnology and Food Sciences, Slovak University of Agriculture in Nitra, Tr. A. Hlinku 2, 94976 Nitra, Slovakia; 5School of Medical and Health Sciences, University of Economics and Human Sciences in Warsaw, Okopowa 59, 010 43 Warsaw, Poland

**Keywords:** kaempferol, sperm cryopreservation, bulls, cryocapacitation, reactive oxygen species, Western blot

## Abstract

Kaempferol (KAE) is a natural flavonoid with powerful reactive oxygen species (ROS) scavenging properties and beneficial effects on ex vivo sperm functionality. In this paper, we studied the ability of KAE to prevent or ameliorate structural, functional or oxidative damage to frozen–thawed bovine spermatozoa. The analysis focused on conventional sperm quality characteristics prior to or following thermoresistance tests, namely the oxidative profile of semen alongside sperm capacitation patterns, and the levels of key proteins involved in capacitation signaling. Semen samples obtained from 30 stud bulls were frozen in the presence of 12.5, 25 or 50 μM KAE and compared to native ejaculates (negative control—Ctrl_N_) as well as semen samples cryopreserved in the absence of KAE (positive control—Ctrl_C_). A significant post-thermoresistance test maintenance of the sperm motility (*p* < 0.001), membrane (*p* < 0.001) and acrosome integrity (*p* < 0.001), mitochondrial activity (*p* < 0.001) and DNA integrity (*p* < 0.001) was observed following supplementation with all KAE doses in comparison to Ctrl_C_. Experimental groups supplemented with all KAE doses presented a significantly lower proportion of prematurely capacitated spermatozoa (*p* < 0.001) when compared with Ctrl_C_. A significant decrease in the levels of the superoxide radical was recorded following administration of 12.5 (*p* < 0.05) and 25 μM KAE (*p* < 0.01). At the same time, supplementation with 25 μM KAE in the cryopreservation medium led to a significant stabilization of the activity of Mg^2+^-ATPase (*p* < 0.05) and Na^+^/K^+^-ATPase (*p* < 0.0001) in comparison to Ctrl_C_. Western blot analysis revealed that supplementation with 25 μM KAE in the cryopreservation medium prevented the loss of the protein kinase A (PKA) and protein kinase C (PKC), which are intricately involved in the process of sperm activation. In conclusion, we may speculate that KAE is particularly efficient in the protection of sperm metabolism during the cryopreservation process through its ability to promote energy synthesis while quenching excessive ROS and to protect enzymes involved in the process of sperm capacitation and hyperactivation. These properties may provide supplementary protection to spermatozoa undergoing the freeze–thaw process.

## 1. Introduction

Long-term storage of spermatozoa enabled by the cryopreservation process is a principal prerequisite for effective artificial insemination (AI) as a critical pillar of livestock breeding worldwide [1]. The use of cryopreserved spermatozoa from genetically superior stud males leads to desirable fertility rates and litter sizes, lower risks for the transmission of diseases within the herd and more cost-effective production of animal products [2,3]. Nevertheless, while sperm freezing, storage and thawing are technically and logistically well-managed procedures particularly in the case of cattle production, the loss of viable spermatozoa during cryopreservation still requires new strategies for the improvement of sperm cryobiology [1,4].

Unlike other cell types, spermatozoa are small cells with a large surface [5] which will affect the intracellular viscosity and transition temperature [2]. Hence, if the cryopreservation procedure is not carried out properly, or if cryoprotective agents are not present in the extender, ice crystals and thermal and osmotic shock may lead to structural sperm damage [1,4], cell death by apoptosis or necrosis [6,7] and reactive oxygen species (ROS) overgeneration, which is considered as the prime mechanism of the loss of sperm function following the freeze–thaw process [7,8]. Accordingly, a high proportion of spermatozoa with peroxidized membranes, oxidative damage to the proteins and DNA [6,9] and high levels of peroxyl radicals and lipid hydroperoxides [8] have been found in frozen–thawed semen samples.

Premature capacitation is yet another observed complication of sperm cryopreservation. Under physiological circumstances, capacitation is initiated by the activation of the transmembrane channels including the cation channels of sperm (CatSper) and the sodium bicarbonate cotransporter (NBC), leading to an influx of calcium (Ca^2+^) and bicarbonate (HCO_3_^−^) into the sperm intracellular compartments, followed by a rise in pH and synthesis of cyclic adenosine monophosphate (cAMP). This will then lead to the activation of protein kinase A (PKA), which will trigger tyrosine kinases, leading to sperm hyperactivation and changes in the architecture of the sperm plasma membrane [7,10]. In the meantime, protein kinase C (PKC) will play essential roles in the acquisition of hyperactive motility as a serine/threonine kinase and a mediator in Ca^2+^ mobilization for the signaling pathways leading to the acrosome reaction [10]. As opposed to the physiological process, cryocapacitation is initiated by the loss of Ca^2+^ and HCO_3_^−^ transmembrane channels responsible for the initiation of sperm activation [7,11]. Prematurely capacitated spermatozoa are characterized by low cAMP production [12] and the loss of cholesterol and phospholipids from the plasma membrane leading to cells with compromised membranes that will exhibit chlortetracycline hydrochloride (CTC) positivity typical for physiologically capacitated spermatozoa [6,11,13]. At the same time, prematurely capacitated spermatozoa will quickly exhaust the energy required for fertilization [6], rendering such compromised semen samples unsuitable for artificial insemination.

Since the pioneering work of Dr. Beconi’s team in the nineties [13,14], several so-called defensive strategies have been developed, which consist in the administration of different supplements to the freezing extenders that would provide an extra layer of cryoprotection to spermatozoa. These may include antifreeze proteins [15] and cytoprotective agents [16], fatty acids [17], antioxidants [18] or energy-rich molecules [19]. In this sense, recent advances have been achieved in the field of natural biomolecules and extracts from medicinal plants and herbs used in ethnopharmacology with a plethora of beneficial effects on the male reproductive system [20,21].

Kaempferol (3,4′,5,7-tetrahydroxyflavone; KAE), which may be found in leeks, broccoli, kale, dill, chives, tea, apples, strawberries and several medicinal plants (*Ginkgo biloba* L., *Aloe vera*, *Rosmarinus officinalis*, etc.), has only recently emerged as a nutraceutical with promising cardioprotective, anticancer, antioxidant, anti-inflammatory, antibacterial and antiangiogenic properties [22,23,24,25]. Recent research has revealed a significant potential of this flavonoid in the protection of spermatozoa exposed to heavy metals or food mutagens, through its mitochondria-stabilizing, antigenotoxic and antioxidant effects [26]. The potential of KAE as a supplement to semen extenders has been previously suggested by Ďuračka et al. [27] who reported a significant maintenance of motion activity and mitochondrial function in extended boar semen. Moreover, our recent research has uncovered that the administration of KAE to a cryopreservation medium led to a significantly lower occurrence of oxidative damage to sperm proteins, lipids and DNA alongside a higher stabilization of the mitochondrial anti-apoptotic Bcl-2 protein. At the same time, the molecule was effective in preventing excessive damage to the sperm heat shock proteins caused by cryopreservation [28]. Nevertheless, the mechanisms underlying the beneficial effects of KAE on male gametes need further understanding.

This study was conceived to assess the impact of KAE on the levels of the ROS types most observed in sperm physiology and pathophysiology in a broader context of changes to structural integrity and functional activity of cryopreserved bovine spermatozoa. Furthermore, we studied the ability of KAE to prevent cryocapacitation through the protein levels of CatSper isoforms 1 and 2 (CatSper1 and CatSper2), NBC, PKA and PKC, as important molecular players in the process of capacitation.

## 2. Results

### 2.1. Conventional Sperm Quality

The sperm motility analysis revealed a significant (*p* < 0.0001) decline in the sperm motion activity of the cryopreserved control (Ctrl_C_) both prior to as well as following the thermoresistance test (Ctrl_C_; Figure 1a) when compared to fresh semen (native control Ctrl_N_). The presence of all KAE doses led to a significant motility stabilization, particularly following the thermoresistance test (*p* < 0.0001) when compared to Ctrl_C_. Significant differences in the motility of spermatozoa challenged by the thermoresistance test were also detected between the native control (Ctrl_N_) and all experimental groups exposed to KAE (*p* < 0.0001 with respect to 12.5 µM and 25 μM KAE; *p* < 0.001 in the case of 50 μM KAE).

Complementary to an improved post–thaw sperm motility, a significantly increased pre- as well as post-thermoresistance test membrane integrity was observed in experimental groups supplemented with 12.5 µM and 25 μM KAE when compared to Ctrl_C_ (*p* < 0.0001; Figure 1b). In comparison with Ctrl_N_, significant post-thermoresistance test differences were observed in all experimental groups administered with KAE (*p* < 0.0001).

Negative effects of cryopreservation were also observed in the case of sperm acrosome integrity. When compared to Ctrl_N_, a significant decrease in the acrosome integrity was observed in Ctrl_C_ (*p* < 0.0001; Figure 1c) as well as in all experimental groups supplemented with KAE (*p* < 0.0001) particularly following the thermoresistance test. Inversely, a significant post-thermoresistance test improvement in the integrity of acrosomal structures was recorded in all KAE-administered experimental groups when compared to Ctrl_C_ (*p* < 0.001).

The negative effects of the freeze–thaw process on the sperm mitochondrial activity were unraveled by the JC-1 assay. Significant differences in the post-thermoresistance test mitochondrial membrane potential were detected among Ctrl_N_ and Ctrl_C_ (*p* < 0.0001; Figure 2a) as well as Ctrl_N_ and all KAE-administered experimental groups (*p* < 0.0001). On the other hand, a significant improvement of the mitochondrial membrane potential following the thermoresistance test was observed in all experimental groups exposed to KAE when compared to Ctrl_C_ (*p* < 0.0001).

An increased proportion of spermatozoa negatively affected by the freeze–thaw process was accompanied by an increased rate of sperm DNA damage, as shown by significant differences in the index of DNA fragmentation among Ctrl_N_ and Ctrl_C_ both prior to as well as following the thermoresistance test (*p* < 0.0001; Figure 2b). The comparative analysis also revealed a significant post-thermoresistance test increase in spermatozoa with fragmented DNA in all experimental groups supplemented with KAE (*p* < 0.0001). Nevertheless, the sperm DNA fragmentation index was significantly lower in all experimental groups when compared to Ctrl_C_ before as well as after the thermoresistance test (*p* < 0.0001).

### 2.2. Capacitation Patterns

The lowest percentage of non-capacitated spermatozoa (“F”-pattern) both prior to as well as following the thermoresistance test was recorded in Ctrl_C_, which was significantly different when compared with Ctrl_N_ (*p* < 0.0001; Figure 3a). A significantly lower proportion of non-capacitated spermatozoa was also recorded in all KAE-administered experimental groups in comparison to Ctrl_N_ (*p* < 0.0001). Inversely, the presence of all KAE doses partially prevented the loss of non-capacitated spermatozoa when compared to the cryopreserved control before as well as after the thermoresistance test (*p* < 0.0001).

Correspondingly, a significantly increased pre- as well as post-thermoresistance test proportion of “B”-pattern spermatozoa, indicating the onset of capacitation, was observed in Ctrl_C_ as well as all KAE-supplemented experimental groups when compared to Ctrl_N_ (*p* < 0.0001; Figure 3b). KAE administration did not fully prevent premature capacitation, as shown by a significantly higher post-thermoresistance test proportion of “B”-pattern spermatozoa in all experimental groups when compared to Ctr_N_ (*p* < 0.0001). Nevertheless, the percentage of capacitated spermatozoa was significantly lower in all KAE-supplemented experimental groups (*p* < 0.0001) when compared to Ctrl_C_.

Along with changes to the acrosome integrity previously observed in the control and experimental groups, a significant increase in “AR”-pattern spermatozoa, indicating a premature acrosome reaction, was observed in Ctrl_C_ in comparison to Ctrl_N_ before as well as after the thermoresistance test (*p* < 0.0001; Figure 3c). On the other hand, administration of 25 μM KAE to the cryopreservation medium resulted in a significantly decreased occurrence of acrosome-reacted spermatozoa in comparison to Ctrl_C_ both prior to as well as following the thermoresistance test (*p* < 0.0001).

### 2.3. Activity of ATP-Ases

The decline in mitochondrial membrane potential was mirrored by a significantly decreased activity of Ca^2+^-ATPase (*p* < 0.05; Figure 4a), Mg^2+^-ATPase (*p* < 0.001; Figure 4b) and Na^+^/K^+^-ATPase (*p* < 0.0001; Figure 4c) in Ctrl_C_ in comparison to Ctrl_N_. A significant decline in the activity of all three ATP-ases was also observed in all KAE-supplemented experimental groups when compared to Ctrl_N_. Nevertheless, the presence of 25 μM KAE led to significant improvement of Mg^2+^-ATPase activity in comparison to Ctrl_C_ (*p* < 0.05). Furthermore, significantly higher activity of Na^+^/K^+^-ATPase was observed in the case of 25 μM KAE (*p* < 0.01) when compared to Ctrl_C_.

### 2.4. Oxidative Profile

As revealed in Figure 5a, global reactive oxygen species (ROS) levels were significantly increased in all cryopreserved groups in comparison to Ctrl_N_ (*p* < 0.001), with Ctrl_C_ presenting the highest ROS production. On the other hand, significantly lower ROS levels were observed in the experimental groups exposed to 12.5 μM KAE (*p* < 0.001) and 25 μM KAE (*p* < 0.0001) in comparison with the cryopreserved control.

In the case of superoxide (O_2_^−^) production, significantly higher levels were observed in Ctrl_C_ (*p* < 0.01; Figure 5b) and the experimental group containing 50 μM KAE in comparison to Ctrl_N_ (*p* < 0.05). On the other hand, significantly lower O_2_^−^ levels were found in the experimental samples cryopreserved in the presence of 25 μM KAE (*p* < 0.05) when compared to the cryopreserved control.

Comparably to O_2_^−^, the highest levels of hydrogen peroxide (H_2_O_2_) were found in Ctrl_C_ as opposed to Ctrl_N_ (*p* < 0.0001; Figure 5c). Significantly higher H_2_O_2_ concentrations were also recorded in all samples supplemented with KAE (*p* < 0.01) when compared to Ctrl_N_. Inversely, H_2_O_2_ concentrations were significantly lower in the experimental samples containing 12.5 µM KAE and 25 µM KAE (*p* < 0.05) in contrast to Ctrl_C_.

Finally, significantly higher levels of hydroxyl radicals (•OH) were recorded in all cryopreserved groups when compared to Ctrl_N_ (*p* < 0.0001; Figure 5d). While KAE was not able to fully prevent •OH overproduction in the experimental groups, significantly lower •OH levels were recorded in all KAE-supplemented groups (*p* < 0.0001 with respect to 12.5 µM KAE and 25 µM KAE; *p* < 0.001 in the case of 50 µM KAE) when compared to Ctrl_C_.

### 2.5. Western Blots

The protein analysis revealed that CatSper1 and CatSper2 levels were negatively impacted by the freeze–thaw process when compared to Ctrl_N_ (Figure 6 and Figure 7a,b). Meanwhile, slightly higher non-significant CatSper1 and CatSper2 protein amounts were recorded in all KAE-supplemented groups in comparison to Ctrl_C_.

As revealed by Figure 6 and Figure 7c, NBC protein levels were significantly reduced following the cryopreservation process, as revealed by differences in its levels among Ctrl_N_ and Ctrl_C_ (*p* < 0.05). A significant reduction in the NBC protein in comparison to fresh semen was also observed in the experimental groups containing 12.5 μM KAE (*p* < 0.05). All KAE doses supplemented to the extender were able to prevent the complete loss of the NBC protein, although no significant differences were noted.

In the case of PKA, significantly lower levels of the protein were observed in the cryopreserved control in comparison to fresh semen (*p* < 0.01; Figure 6 and Figure 7d). Nevertheless, a significantly higher PKA protein amount was recorded in the experimental group supplemented with 25 μM KAE (*p* < 0.05) when compared to Ctrl_C_.

A similar phenomenon was observed in the case of the PKC protein (Figure 6 and Figure 7d), with its levels being significantly decreased in Ctrl_C_ as opposed to Ctrl_N_ (*p* < 0.05). In the meantime, administration of 25 μM KAE led to a stabilization of the PKC protein against cryodamage when compared to Ctrl_C_ (*p* < 0.05).

## 3. Discussion

The administration of molecules with the ability to offer an additional layer of protection to the sperm architecture and function challenged with thermal, physio-chemical, osmotic or oxidative stress during cryopreservation has become a popular trend in animal breeding. Since the first reports suggesting that vitamin C and vitamin E, besides acting as effective ROS-quenchers, may also stabilize the plasma membrane of cryopreserved bovine spermatozoa against premature capacitation [13,14], a vast array of reports has emerged over the past decade, suggesting that plant-based bioactive compounds could improve the post-thaw semen quality even more efficiently as conventional antioxidants [20,21,29]. In our case, the experimental rationale was built upon current evidence on the biological effects of KAE on male reproductive tissues and cells, with the aim of understanding the specific mechanisms by which this biomolecule exerts its protective and/or attenuating properties against cryogenic shock-induced sperm deterioration.

The broad family of flavonoids comprises highly efficient ROS-scavengers and metal-chelating agents, which may be attributed to their peculiar chemistry [30]. The basic chemical structure of KAE is made up of a double bond at C2–C3; an oxo group at C4; and hydroxyl groups at C3, C5, and C4 which may easily interact with unpaired oxygen atoms. Once ROS bind to the flavonoid structure, a stable compound is formed, preventing further oxidation reactions from occurring [31]. At the same time, it has been proposed that KAE slows down the oxidation of α-tocopherol and promotes the activity of glutathione-S-transferase, UDP-glucuronosyltransferase and NAD(P)H-quinone oxidoreductase [32], thus also acting as an efficient secondary antioxidant, assisting in the preservation of a stable intracellular oxidative balance.

The inherent lipophilic nature of KAE, coupled with a high affinity to quench and dispose of ROS primarily attacking the membrane lipids, may serve as an explanation for a notable stabilization of the membrane integrity along with the prevention of premature capacitation revealed in this study as well as in earlier reports on boar [27], bull [28] and buffalo sperm [33]. In this sense, we may assume that similarly to other antioxidant molecules [14], KAE will primarily interact with the sperm plasma membrane, leading to its stabilization and protection against excessive lipid peroxidation [28]. Consequently, there will be a lower risk for the loss of the membrane’s inherent semi-permeability, the membrane’s ability to maintain a desirable intracellular milieu and the capacity of spermatozoa to undergo capacitation changes [34,35]. Moreover, KAE seems to exert its protective effects on the acrosomal vesicle, which is necessary for the sperm to reach the zona pellucida and accomplish fertilization [35].

Nevertheless, our data collection coupled with previously published observations [27,28,33,36] indicates that the stabilization of the sperm mitochondrial apparatus may also represent an important mechanism by which KAE preserves post-thaw sperm vitality. Mitochondria, regarded as “sperm powerhouses”, are highly prone to suffer injuries during freezing, and any abnormalities in their morphology or function are reflected in alterations to ATP synthesis, Ca^2+^ homeostasis, ROS overproduction and apoptosis, all of which may contribute to low sperm quality [37]. A notable preservation of the sperm mitochondrial membrane potential alongside the activities of Mg^2+^-ATPase and Na^+^/K^+^-ATPase in the presence of KAE complements earlier reports observing a stabilization of mitochondrial succinate dehydrogenase in chilled boar semen exposed to 10–25 µM KAE [27]. Other authors have also hypothesized that KAE presents the ability to modulate the mitochondrial Ca^2+^ uniporter without being dependent upon ATP [36]. All in all, we may speculate that KAE could act as a protective agent for the mitochondrial membranes on one hand and a ROS-quencher on the other, which may result in a higher percentage of spermatozoa with a functional mitochondrial metabolism and desirable motility rates as previously observed in animals as well as humans [26,27,28,33].

An evaluation of the ability of KAE to prevent the overgeneration of specific ROS types has revealed that this flavonoid was particularly effective in the case of •OH, which is directly involved in lipid peroxidation [38], oxidative DNA damage and apoptosis [39]. This observation may be directly linked to the inherent chemical structure of KAE for binding to and disposing of already generated ROS, as well as its potential to stabilize the mitochondrial function as the primary source of ROS in semen, decreasing the levels of O_2_^−^ and H_2_O_2_ that could interact together to produce •OH through the Fenton and Haber–Weiss reaction [40]. We may speculate that rather than eliminating superoxide and hydrogen peroxide, KAE strives to control both free radicals in their physiologically relevant levels that are crucial for a successful capacitation and acrosome reaction. This hypothesis could be supported by earlier reports on bull [28], buffalo [33] and human [26] spermatozoa observing a notable antioxidant potential of KAE translated into a stabilized antioxidant capacity and superoxide dismutase activity, accompanied by a decline in lipid peroxidation that is caused predominantly by •OH. The assumption that both the plasma membrane as well as mitochondria could benefit from KAE may be fortified by our earlier study revealing that the administration of 12.5 and 25 μM KAE during bovine sperm cryopreservation led to a more desirable mitochondrial BAX/Bcl-2 ratio accompanied by the stabilization of heat shock proteins 70 and 90 located on the sperm surface [28].

Our hypothesis that KAE could be effective in the prevention of premature sperm activation was furthermore validated by Western blot analysis, indicating that supplementation with 25 μM KAE could prevent the inactivation of protein kinases A and C either due to thermal or oxidative stress [41]. This may be explained by the ability of KAE to dispose of ROS before they can reach the axoneme and outer dense fibers involved in the process of sperm hyperactivation by PKA [42] while stabilizing the cytoplasmic and acrosomal membranes for a PKC-mediated acrosome reaction to occur when needed [43,44]. PKA and PKC stabilization in KAE-supplemented experimental groups could also be explained by a higher degree of protection of the sperm membrane containing transmembrane channels needed for the activation of adenylyl cyclase (ADCY) and cyclic adenosine monophosphate (cAMP) synthesis that will trigger the kinases. This hypothesis could be also supported by earlier reports observing that polyphenols and flavonoids may modulate AMP-activated protein kinase and mitogen-activated protein kinase activity and interfere with cAMP-activated protein kinase, guanylate cyclase and ADCY signaling in different cell types [45,46,47]. On the other hand, the effects of KAE on the CatSper1, CatSper2 and NBC levels were insignificant, although a notable post-thaw improvement was recorded in the case of CatSper2 (in the presence of 12.5 and 25 μM KAE) and NBC (in the presence of 50 μM KAE). This may be explained by the lipophilic nature of KAE enabling the molecule to be incorporated into the plasma membrane and thus at least partially stabilize the network of transmembrane channels against oxidative insults. While the CatSper channel is involved in membrane depolarization and subsequent Ca^2+^ influx essential for flagellar movement, NBC is responsible for Na^+^ and HCO_3_^−^ transport, cytoplasmatic alkalization, membrane hyperpolarization and subsequent activation of the cAMP/PKA pathway [7]. While we may speculate that cryo-induced damage to the sperm membrane may have led to the loss of such channels, increased PKA amounts alongside a stabilization of the membrane-bound ATP-ases suggests that KAE exhibits at least a partial protection of the CatSper- and NBC-initiated capacitation pathway. Nevertheless, it is known that other ion channels also play critical roles in sperm capacitation, such as SLC26 transporters responsible for HCO_3_^−^/Cl^−^ regulation or the sperm Na^+^/H^+^ exchanger (sNHE). Since KAE has been previously shown to modulate the activity of other potassium and chloride channels in different cell lines [42,43,48,49], it would be feasible to study the response of other transmembrane transporters involved in sperm activation to the presence of KAE in the future.

An important limitation of this study lies in the sample type. Ejaculates for this study were obtained from highly fertile Holstein stud bulls, bred for the purposes of insemination doses. Such semen samples traditionally present a supreme quality and a higher tolerance to cryodamage [50]. These molecular properties, coupled with the technologically well-managed and automated cryopreservation procedure used in this study, may lie behind a relatively small improvement of several sperm quality characteristics in the presence of KAE in comparison with the cryopreserved control. Bull semen quality depends on a wide array of factors, including age, season and breed [50,51]. At the same time, different cryopreservation protocols are available for breeders, from a traditional slow and/or manual freezing procedure to a computer-assisted programmed cryopreservation or vitrification [52,53]. Therefore, we may speculate that more prominent protective effects of KAE might have been observed in the case of ejaculates obtained from older breeding bulls or beef breeds diluted and cryopreserved using different freezing techniques.

Finally, while our collected data are promising, we must keep in mind a proper concentration range of the flavonoid for it to exert its most beneficial effects. As seen in our experiments, the most efficient dose selected was 25 μM KAE. Despite a certain degree of positive influence of 50 μM KAE being observed when compared to the untreated cryopreserved control, the “double-edged sword” properties of KAE became noticeable at this dose. Therefore, we may suggest that the precise effects of KAE on sperm vitality may primarily depend on the dosage since its higher concentrations seem to act in a counterproductive manner, which corroborates a previous hypothesis that high amounts of KAE may cause self-oxidation [54]. If applied in excess, the biomolecule may exert pro-oxidant activities coupled with a high affinity to nuclear acids, leading to a compromised DNA condensation and a higher incidence of single- as well as double-strand breaks [55]. Taking this discrepancy into account, we must emphasize the importance of selecting the appropriate KAE dose for cellular treatment with caution.

## 4. Materials and Methods

### 4.1. Collection and Cryopreservation of Ejaculates

Ejaculates were obtained from 30 stud Holstein–Friesian bulls (Slovak Biological Services, a.s., Nitra, Slovakia) using an artificial vagina and moved to the laboratory in a Mini Bio Isotherm system (M&G Int., Renate, Italy).

Each sample was divided into five aliquots of equal volume. The first one, representing the negative control (fresh semen), was diluted in Dulbecco’s phosphate-buffered saline (DPBS) (without calcium and magnesium; Sigma-Aldrich, St. Louis, MO, USA) to 44 million sperm/mL and immediately assessed. The residual aliquots were diluted with a cryopreservation medium that consisted of Triladyl (Minitub GmbH, Tiefenbach, Germany), 20% (*w*/*v*) fresh egg yolk, glycerol, Tris, citrate, saccharides, buffers and distilled water. The medium was furthermore fortified with 500 µg streptomycin/mL final dilution (Sigma-Aldrich, St. Louis, MO, USA) and 500 IU penicillin/mL final dilution (Sigma-Aldrich, St. Louis, MO, USA) [56]. For the positive (cryopreserved) control, 0.5% DMSO (dimethyl sulfoxide; Sigma-Aldrich, St. Louis, MO, USA) was added to the extender, while the extender for the experimental groups was administered with 12.5, 25 or 50 μM KAE (Sigma-Aldrich, St. Louis, MO, USA) that was dissolved in DMSO beforehand. The optimal concentration range for KAE was decided based on our previous experiments on boar spermatozoa [27] and subsequently confirmed on bovine specimens [28]. The samples were then loaded into 250 µL French straws (11 million sperm/straw) and frozen with the Digitcool 5300 ZB 250 digital freezer (IMV, L’Aigle, France) [6,11]. Cryopreserved specimens were stored in liquid nitrogen (−196 °C) for 6 weeks.

For the analyses, frozen samples were thawed in a water bath (37 °C) for 90 s, washed 3 times with DPBS and centrifuged at 300× *g* for 10 min.

Conventional sperm quality characteristics and sperm capacitation patterns were assessed twice in each cryopreserved sample:Immediately after thawing (pre-thermoresistance test).Following a thermoresistance test (post-thermoresistance test): The samples were distributed into test tubes with support grids and placed in a water bath (37 °C) for 2 h. After that, an aliquot of each sample was withdrawn for evaluation [57].

### 4.2. Conventional Sperm Quality Assessment

Computer-aided sperm analysis (CASA; Version 14.0 TOX IVOS II.; Hamilton-Thorne Biosciences, Beverly, MA, USA) was used to assess sperm motility. Ten microscopic fields were subjected to each round of analysis, with the capture of at least 300 cells [11].

Sperm membrane integrity was evaluated with a double fluorescent staining protocol comprising CFDA (carboxyfluorescein diacetate; Sigma-Aldrich, St. Louis, MO, USA) and DAPI (4′,6-diamidino-2-phenylindole; Sigma-Aldrich, St. Louis, MO, USA) as previously described [6,10]. The samples were measured in a 96-well microplate with the combined spectro-fluoro-luminometer Glomax Multi^+^ (Promega, Madison, WI, USA) [11].

Acrosome integrity was analyzed by a Peanut agglutinin (PNA; FITC conjugate; Sigma-Aldrich, St. Louis, MO, USA; 10 μmol/L in DPBS) and DAPI double staining protocol as previously described [6,11]. The fluorescent signal emitted by the cells was captured by the Glomax Multi^+^ combined spectro-fluoro-luminometer [11].

The sperm mitochondrial membrane potential was evaluated with the JC-1 assay (Cayman Chemical, Ann Arbor, MI, USA) as previously published [6,10]. All stained specimens were transferred to a dark 96-well plate and analyzed using the Glomax Multi^+^ combined spectro-fluoro-luminometer [11].

The sperm DNA fragmentation index was assessed using the Halomax commercial kit (Halotech DNA, Madrid, Spain) according to the instructions by the manufacturer, and a minimum of 300 processed spermatozoa per sample were counted using an epifluorescence microscope with a ×40 magnification (Leica Microsystems, Wetzlar, Germany) [6,11].

### 4.3. Capacitation Status

The capacitation patterns of spermatozoa in each sample were analyzed using the chlortetracycline (CTC) staining method as previously published [6,11]. At least 200 sperm per sample were categorized into three groups according to the CTC signal: “F” pattern (non-capacitated); “B” pattern (capacitated); “AR” pattern (acrosome-reacted) [6,11]. An epifluorescence microscope with a ×40 magnification (Leica Microsystems, Wetzlar, Germany) was used for the evaluation.

### 4.4. Activity of ATP-Ases

The activities of Ca^2+^-ATPase, Mg^2+^-ATPase and Na^+^/K^+^-ATPase were quantified according to the protocol published by Zhao and Buhr [58]. Each sample was diluted with HEPES using a 1:3 ratio and centrifuged (500× *g*, 10 min) through 35% Percoll (Sigma-Aldrich, St. Louis, MO, USA). The resulting pellet was washed in HEPES, diluted in DPBS and homogenized on ice thrice [58].

Ca^2+^-ATPase activity was measured with 80 mM NaCl (Centralchem, Bratislava, Slovakia), 20 mM KCl (Centralchem, Bratislava, Slovakia), 0.2 mM CaCl_2_ (Centralchem, Bratislava, Slovakia) and 2 mM Ouabain (Sigma-Aldrich, St. Louis, MO, USA). Mg^2+^-ATPase activity was assessed using 80 mM NaCl, 20 mM KCl, 6 mM MgCl_2_ (Centralchem, Bratislava, Slovakia), 1.5 mM egtazic acid (Sigma-Aldrich, St. Louis, MO, USA) and 2 mM Ouabain. Na^+^/K^+^-ATPase activity was quantified using 80 mM NaCl, 20 mM KCl, 6 mM MgCl_2_, 10 mM NaF (Centralchem, Bratislava, Slovakia) and 2 mM Ouabain [58,59].

Each pre-treated sample was pre-incubated at 37 °C for 5 min, and the reaction was initiated by adding 3 mM ATP (Sigma-Aldrich, St. Louis, MO, USA) and incubated at 37 °C for 15 min. Each reaction was terminated with 10 mM HgCl (Centralchem, Bratislava, Slovakia) and kept on ice for 10 min. The ATP-ase activity was measured at 820 nm using a Cary UV-VIS spectrophotometer (Cary Systems, Santa Clara, CA, USA) and determined as phosphate release [60].

### 4.5. Oxidative Profile

Global ROS production comprising both extra- as well as intracellular ROS was assessed by luminol (5-amino-2,3-dihydro-1,4-phthalazinedione; Sigma-Aldrich, St. Louis, MO, USA)-based chemiluminescence [6,11]. The chemiluminescent signal was measured for 15 min using the Glomax Multi^+^ combined spectro-fluoro-luminometer.

Superoxide production was evaluated with the colorimetric nitroblue-tetrazolium (NBT) test as previously described [11]. The optical density of the reaction was measured at a wavelength of 620 nm against 570 nm using the Glomax microplate photometer (Promega, Madison, WI, USA). The data were expressed as a percentage of the control (100%).

Hydrogen peroxide production was quantified with the fluorescent Amplex^®^ Red reagent (Thermo Fisher Scientific; Waltham, MA, USA) and monitored with the Glomax Multi^+^ combined spectro-fluoro-luminometer as previously published [6,11].

Fluorescent aminophenyl fluorescein (APF; Thermo Fisher Scientific; Waltham, MA, USA) was used to evaluate hydroxyl radical levels. The intensity of APF fluorescence was captured with the Glomax Multi^+^ combined spectro-fluoro-luminometer as previously described [6,11].

### 4.6. Western Blots

For the Western blots, spermatozoa were washed out of the suspensions with a single-layer Percoll Plus (Sigma-Aldrich, St. Louis, MO, USA) density gradient [11]. The cells were then lysed with RIPA buffer (Sigma-Aldrich, St. Louis, MO, USA)/protease inhibitor (Sigma-Aldrich, St. Louis, MO, USA) overnight. The samples were centrifuged at 11,828× *g* for 10 min at 4 °C, and the resulting supernatants were subjected to the determination of proteins using a commercially available total protein kit (DiaSys, Holzheim, Germany) [7,11,28].

The lysates were processed with 4× Laemli buffer (BioRad, Hercules, CA, USA) and β-mercaptoethanol (Sigma-Aldrich, St. Louis, MO, USA) and boiled (95 °C, 10 min). The samples were loaded into Mini-PROTEAN TGX polyacrylamide gels (BioRad, Hercules, CA, USA), which were subjected to electrophoresis (90 V, 2 h), and visualized with the ChemiDoc Imaging System (BioRad, Hercules, CA, USA) to confirm the loading uniformity. The gels were then transferred to PVDF membranes (Trans-Blot Turbo Pack; BioRad, Hercules, CA, USA) using the Trans-Blot Turbo Transfer System (BioRad, Hercules, CA, USA). The resulting membranes were washed for 3 × 10 min in Tris-buffered saline (TBS; BioRad, Hercules, CA, USA) and stained with Ponceau S solution (Sigma-Aldrich, St. Louis, MO, USA) to visualize the bands. Subsequently, the membranes were cut into smaller pieces carrying the protein of interest. The pieces were then washed with TBS (3 × 10 min) and blocked with 5% skim milk (Blotting grade blocker; BioRad, Hercules, CA, USA) in TBS containing 0.1% Tween-20 (Sigma-Aldrich, St. Louis, MO, USA) for 2 h. Finally, the membranes were incubated with one of the primary antibodies (dilution 1:1 000 in 5% milk/TBS/0.1% Tween-20) overnight: CATSPER1 Polyclonal Antibody (#PA5-75788; Invitrogen, Waltham, MA, USA); CATSPER2 Polyclonal Antibody (#PA5-41072; Invitrogen, Waltham, MA, USA); Anti-Na^+^/HCO_3_^−^ Contransporter Polyclonal Antibody (#AB3212-I; EMD Milipore Corporation; Temecula, CA, USA); PKA alpha Antibody (#PA5-17626; Invitrogen, Waltham, MA, USA); PKC alpha Polyclonal Antibody (#PA5-17551; Invitrogen, Waltham, MA, USA); beta Actin Polyclonal Antibody (#PA1-46296; Invitrogen, Waltham, MA, USA).

Following an overnight primary incubation, the membranes were washed in 1% milk in TBS/0.2% Tween-20 for 5 × 10 min and treated with a secondary anti-rabbit antibody (GE Healthcare, Chicago, IL, USA) diluted 1:15,000 in 1% milk in TBS/0.2% Tween-20 for 1 h. The membranes were then washed in TBS/0.2% Tween-20 for 5 × 10 min. Finally, the membranes were treated with the ECL substrate (GE Healthcare, Chicago, IL, USA) and visualized with the ChemiDoc Imaging System [9,10,30]. Protein levels were evaluated with the BioRad Image Lab Software 6.1 (BioRad, Hercules, CA, USA) through image acquisition and densitometric analysis. This software interprets raw data in three dimensions with the length and width of each band such that the chemiluminescent signal emitted from the blot is registered in the third dimension as a peak rising out of the blot surface. The density of each band was measured as the total volume under the three-dimensional peak, which could be viewed in two dimensions to adjust the precise width of the band to account for the area under the peak of interest. The Band Analysis tool was used to determine the density of the bands in all blots [61].

### 4.7. Statistics

Statistical analysis was carried out using the GraphPad Prism program (version 10.1.1 for Mac; GraphPad Software, La Jolla, CA, USA). For the sperm quality parameters assessed prior to and following the thermoresistance test, two-way ANOVA and Tukey’s post hoc test were selected. The remaining sperm characteristics were processed using one-way ANOVA with Greenhouse–Geisser correction and Tukey’s post hoc test. The level of significance was set at **** *p* < 0.0001, *** *p* < 0.001, ** *p* < 0.01 and * *p* < 0.05. We focused on the following comparisons:−The native control (fresh semen; Ctrl_N_) was compared to the cryopreserved control (Ctrl_C_);−The experimental groups were compared to both controls.

## 5. Conclusions

To conclude, we may suggest that KAE seems to exert its effects primarily in the sperm mitochondria, disposing of ROS involved in oxidative chain reactions and thus stabilizing the structural integrity of the male gamete and the signaling machinery responsible for proper sperm activation to occur when necessary. Nevertheless, several additional aspects should be taken into consideration before definitive conclusions can be reached. Since the capacitation pathways are complex and depend on a broad array of molecules, it will be of importance to assess the impact of KAE on other membrane ion transporters as well as downstream enzymes responsible for the activation of hyperactivated sperm motility and changes to the membrane in preparation for the acrosome reaction. Finally, the effect of KAE on the fertilization ability of frozen–thawed bovine spermatozoa needs to be verified, either through artificial insemination or in vitro fertilization.

## Figures and Tables

**Figure 1 ijms-25-04129-f001:**
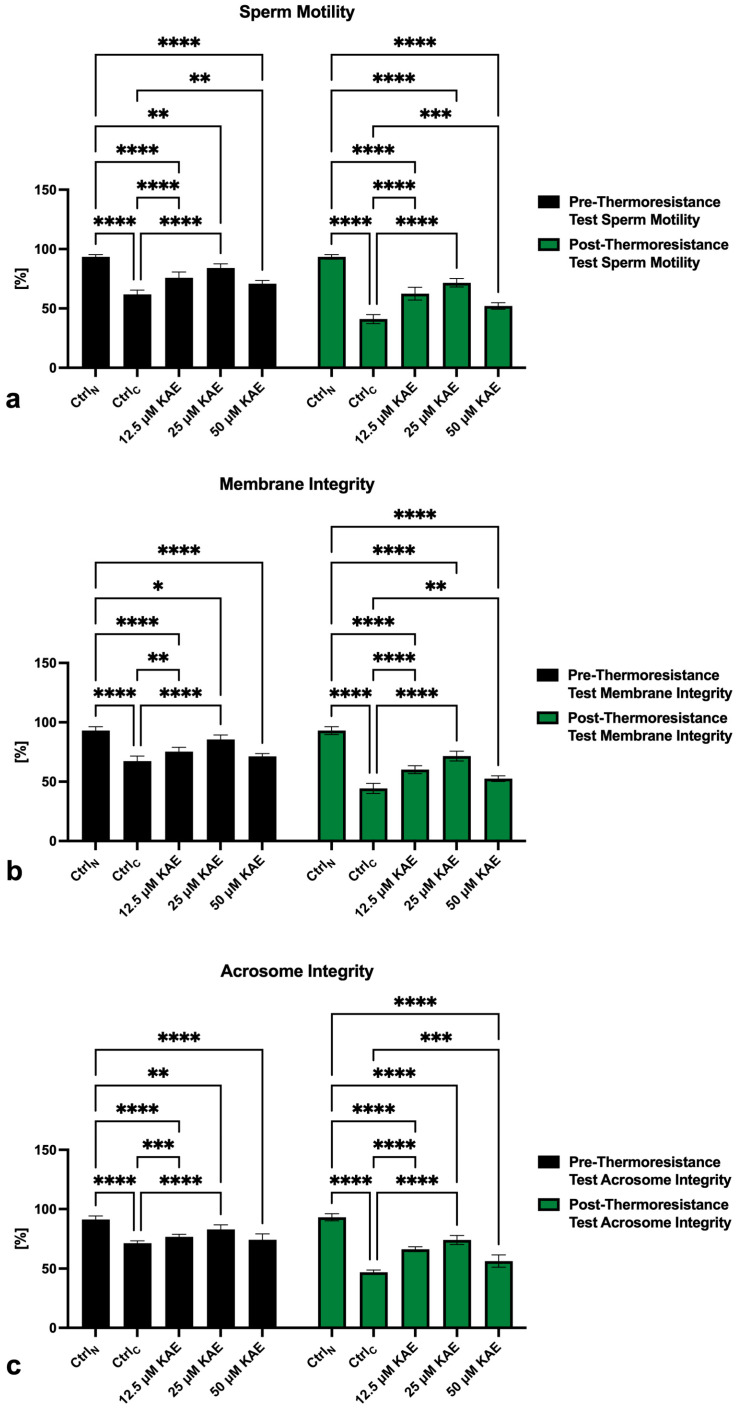
Pre- and post-thermoresistance test motility (**a**), membrane integrity (**b**) and acrosome integrity (**c**) of bovine spermatozoa (*n* = 30) in fresh state (negative control; Ctrl_N_) and cryopreserved in the absence (positive control; Ctrl_C_) or presence of different kaempferol (KAE) concentrations. Each experiment was carried out in triplicate. Mean ± S.D. * *p* < 0.05; ** *p* < 0.01; *** *p* < 0.001; **** *p* < 0.0001.

**Figure 2 ijms-25-04129-f002:**
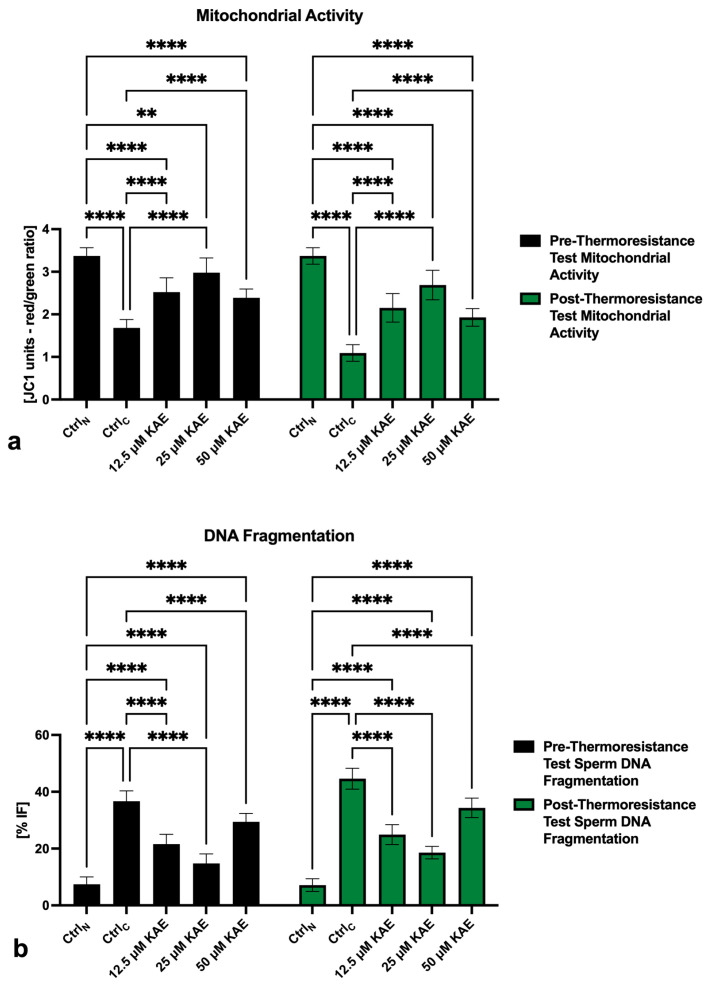
Pre- and post-thermoresistance test mitochondrial activity (**a**) and DNA fragmentation (**b**) of bovine spermatozoa (*n* = 30) in fresh state (negative control; Ctrl_N_) and cryopreserved in the absence (positive control; Ctrl_C_) or presence of different kaempferol (KAE) concentrations. Each experiment was carried out in triplicate. Mean ± S.D. ** *p* < 0.01; **** *p* < 0.0001.

**Figure 3 ijms-25-04129-f003:**
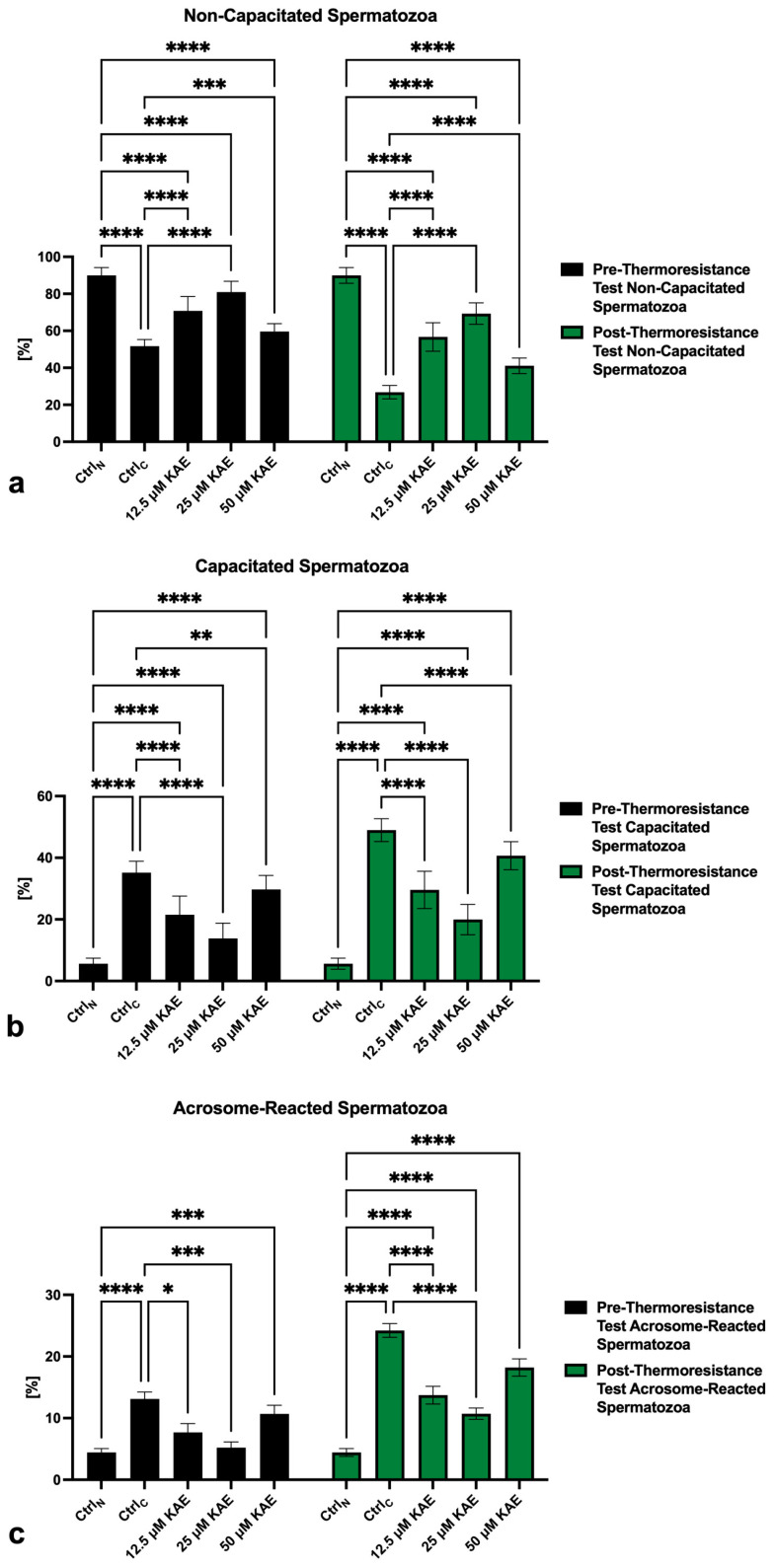
Pre- and post-thermoresistance test proportion of non-capacitated (**a**), capacitated (**b**) and acrosome-reacted (**c**) bovine spermatozoa (*n* = 30) in fresh state (negative control; Ctrl_N_) and cryopreserved in the absence (positive control; Ctrl_C_) or presence of different kaempferol (KAE) concentrations. Each experiment was carried out in triplicate. Mean ± S.D. * *p* < 0.05; ** *p* < 0.01; *** *p* < 0.001; **** *p* < 0.0001.

**Figure 4 ijms-25-04129-f004:**
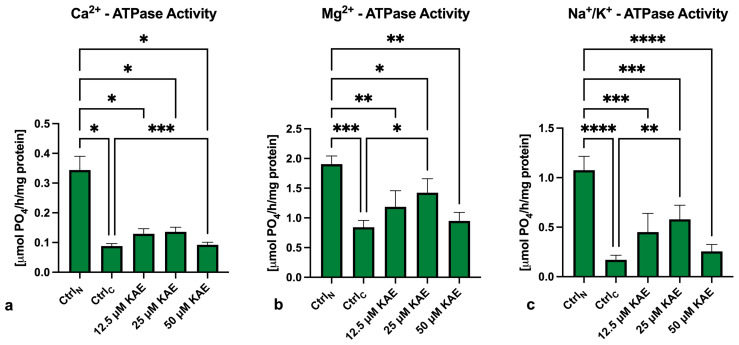
Activity of Ca^2+^-ATP-ase (**a**), Mg^2+^-ATP-ase (**b**) and Na^+^/K^+^-ATP-ase (**c**) of bovine spermatozoa (*n* = 30) in fresh state (negative control; Ctrl_N_) and cryopreserved in the absence (positive control; Ctrl_C_) or presence of different kaempferol (KAE) concentrations. Each experiment was carried out in triplicate. Mean ± S.D. * *p* < 0.05; ** *p* < 0.01; *** *p* < 0.001; **** *p* < 0.0001.

**Figure 5 ijms-25-04129-f005:**
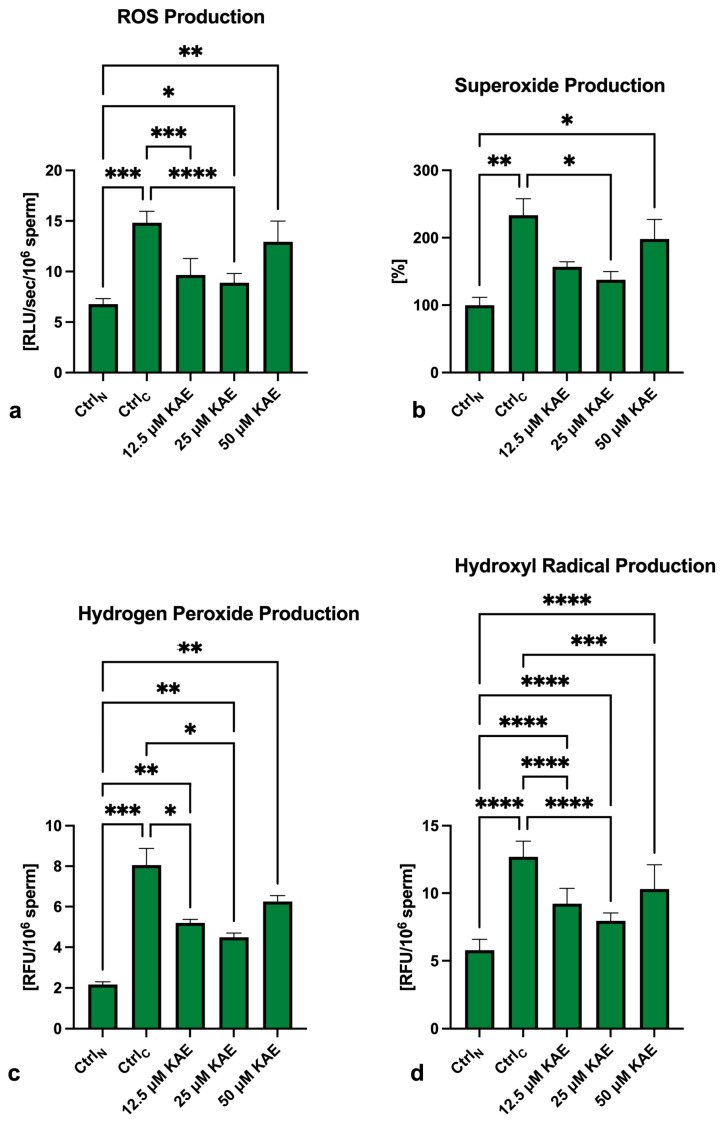
Reactive oxygen species (ROS) (**a**), superoxide (**b**), hydrogen peroxide (**c**) and hydroxyl radical (**d**) production by bovine spermatozoa (*n* = 30) in fresh state (negative control; Ctrl_N_) and cryopreserved in the absence (positive control; Ctrl_C_) or presence of different kaempferol (KAE) concentrations. Each experiment was carried out in triplicate. Mean ± S.D. * *p* < 0.05; ** *p* < 0.01; *** *p* < 0.001; **** *p* < 0.0001.

**Figure 6 ijms-25-04129-f006:**
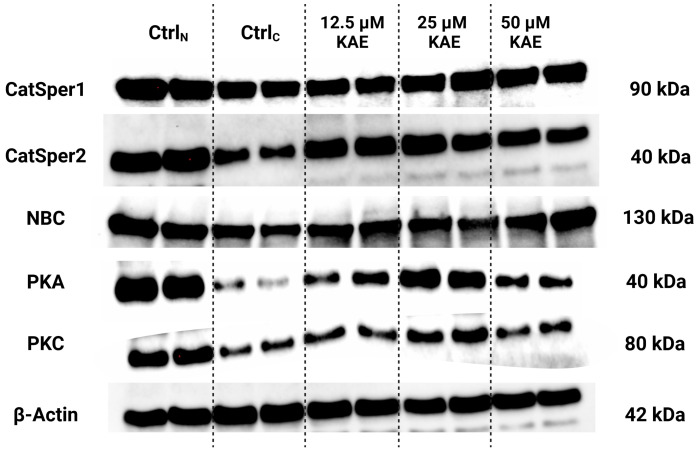
Protein amounts of the cation channels of sperm isoforms 1 and 2 (CatSper1 and CatSper2), sodium bicarbonate cotransporter (NBC), protein kinase A (PKA), protein kinase C (PKC) and β-Actin (housekeeping protein) in bovine spermatozoa in fresh state and cryopreserved in the absence or presence of different kaempferol (KAE) concentrations, as determined by Western blotting. Original photos of the gels, membranes and blots are available as Appendix A. Created with (Supplementary: Confirmation of Publication and Licensing Rights) BioRender.com (accessed on 23 December 2023).

**Figure 7 ijms-25-04129-f007:**
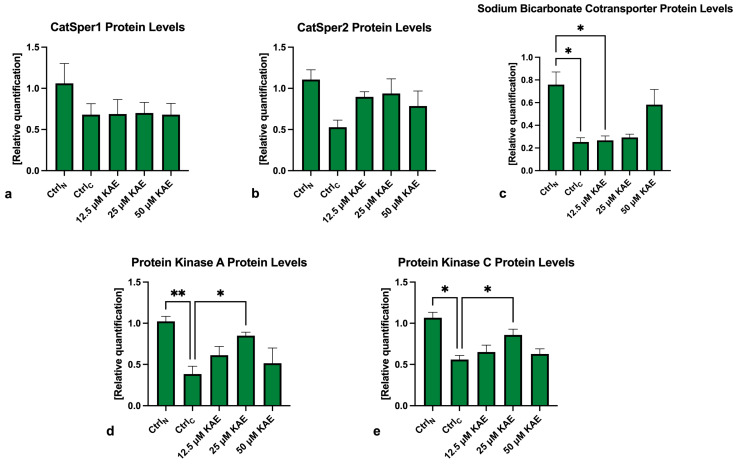
Graphical representation of the relative quantification of the CatSper1 (**a**), CatSper2 (**b**), NBC (**c**), PKA (**d**) and PKC (**e**) protein in bovine spermatozoa (*n* = 30) in fresh state (negative control; Ctrl_N_) and cryopreserved in the absence (positive control; Ctrl_C_) or presence of different kaempferol (KAE) concentrations. Each experiment was carried out in triplicate. Mean ± S.D. * *p* < 0.05; ** *p* < 0.01.

## Data Availability

The data presented in this study are available upon reasonable request from the corresponding author.

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
