# Peer review of "Kaempferol as an Alternative Cryosupplement for Bovine Spermatozoa: Cytoprotective and Membrane-Stabilizing Effects"

_ijms, 2024, doi:10.3390/ijms25074129_

Round 1
Reviewer 1 Report
Comments and Suggestions for Authors
Comments to authors
The study dealing with “Kaempferol as an Alternative Cryosupplement to Bovine Spermatozoa: Cytoprotective, Antibacterial and Membrane-Stabilizing Effects” is an interesting subject. However, in your draft manuscript, more descriptions of the methods are necessary. In addition, the figures need to be modified for better understanding. Therefore, I think it is necessary to revise the paper overall. The detailed major comments are provided below.
Major comments:
1) A word once abbreviated must only be used as an abbreviation from the second use (e.g. line 103, kaempferol).
2) In introduction part, the authors mentioned that CatSper, NBC, PKA, and PKC play key roles in capacitation (line 108-110). Please explain it in detail.
3) I think it would be good to unify the terminology in tables and main texts. For example, acrosome integrity or stability? (table 1 and line 131)
4) Please check overall typographical errors. (e.g. line 154, 184 and so on).
5) In line 221-223, Despite no significant differences, authors should clarify how they concluded that KAE gives certain protection to CatSper1 during cryopreservation.
6) In line 357, the authors need to provide additional explanation about the roles of PKA and PKC.
7) In line 374, please add discussion of results for CatSper1, CatSper2, and NBC.
8) In M&M part, how did authors set KAE concentration? Please explain the reason with reference.
9) Please describe in detailly why sperm motility, membrane integrity, acrosome integrity, membrane potential and so on... are important during cryopreservation in discussion part.
10) Authors should explain how many experiments were repeated (n) for this study in M&M part.
Comments on the Quality of English LanguageN/A
Author Response
Reviewer 1
- A word once abbreviated must only be used as an abbreviation from the second use (e.g. line 103, kaempferol).
Thank you for this remark. The manuscript has been revised accordingly and abbreviations used where necessary.
- In introduction part, the authors mentioned that CatSper, NBC, PKA, and PKC play key roles in capacitation (line 108-110). Please explain it in detail.
Thank you for this comment. More details have been provided with respect to the proteins of interest.
- I think it would be good to unify the terminology in tables and main texts. For example, acrosome integrity or stability? (table 1 and line 131)
The terminology has been unified throughout the manuscript. Thank you.
- Please check overall typographical errors. (e.g. line 154, 184 and so on).
Thank you for this remark. The manuscript has been checked for typos and corrected where necessary.
- In line 221-223, Despite no significant differences, authors should clarify how they concluded that KAE gives certain protection to CatSper1 during cryopreservation.
Thank you for this comment. We have provided more clarification to the hypothesis.
- In line 357, the authors need to provide additional explanation about the roles of PKA and PKC.
Thank you for this comment. More details have been provided with respect to the proteins of interest.
- In line 374, please add discussion of results for CatSper1, CatSper2, and NBC.
Thank you for this comment. More details have been provided with respect to the proteins in the Discussion section.
- In M&M part, how did authors set KAE concentration? Please explain the reason with reference.
The optimal concentration range for KAE was decided upon our previous experiments on boar spermatozoa, and subsequently confirmed on bovine specimens. Relevant references have been added. Thank you.
- Please describe in details why sperm motility, membrane integrity, acrosome integrity, membrane potential and so on... are important during cryopreservation in discussion part.
Thank you for this comment. More details have been provided to each conventional sperm quality parameter in the Discussion section.
- Authors should explain how many experiments were repeated (n) for this study in M&M part.
Thank you for this recommendation. The number of repettions has been added to the manuscript.
Reviewer 2 Report
Comments and Suggestions for Authors
Banas et al. studied the effects of Kaempferol (KAE) as an alternative cryo-supplement to bovine spermatozoa. The findings are not striking, supporting using KAE as a cryoprotectant of bull spermatozoa during the freezing-thawing process.
Although statistically significant, the results in the tables demonstrate that they have limited clinical relevance that merits the incorporation of KAE in the semen extenders.
No functional studies compared the fertilizing ability to demonstrate whether the KAE treatment is a good addition to the cryopreservation semen extender. For instance, incubation of frozen-thawed spermatozoa at 37C for two hours (Thermoresistance test) to determine sperm viability and evaluation of sperm capacitation and acrosome reaction should be presented.
The fact that there is less protein in immunoblots from frozen-thawed spermatozoa compared to fresh or treated ones doesn't mean that the antioxidant treatment improves the expression of proteins. Instead, there is a prevention of losing these proteins because of membrane damage during frozen-thawing. It is more likely that KAE prevents deterioration of the plasma membrane (e.g. decreasing lipid peroxidation) and, thus, avoids the leaking of proteins from the spermatozoon.
The rationale to include the evaluation of bacterial profiles should be stated. This study seems to merit another complete research rather than being a small part of this manuscript.
The discussion is a mere repetition of results, and no molecular mechanisms are described to support the role of KAE as a protector of spermatozoa during cryopreservation.
Specific comments:
- L80: Ref 16 does not talk about bovine cryopreservation. Please change it for a specific review on bovine semen (PMID: 37001221). Also, pioneer work on antioxidants and bovine cryopreservation and the effects of antioxidants on bull sperm function (motility and capacitation) by Dr. Beconi's group has been omitted in the introduction and discussion.
- It is easier for the reader to see the data as bar graphs instead of tables. So many numbers and variables make the analysis very difficult. For instance, It would be more illustrative to clearly present the motility, membrane and acrosome integrity and mitochondrial activity as bar graphs to show differences among samples in Table 1. For these parameters, particularly for sperm motility and acrosome integrity, the differences between the frozen-thawed spermatozoa control or treated with KAE are not clinically relevant (differences less than 30%).
- It is unclear why the effect of 50 uM KAE to protect DNA integrity is lower than that of the other concentrations. How do you explain this decrease in protection?
- The KAE treatment does not lower premature sperm capacitation and ROS levels in spermatozoa.
- Western blots are not well presented, as it is difficult to see the molecular mass of the bands showing. The blots are cut; thus, it is not possible to see if other bands are also present (like for CatSper 2 or beta-actin). These two proteins are not found as doublets; therefore, these blots suggest a problem with antibody specificity and immunoblotting procedure. Was the same band stripped to blot for all proteins? The background is not even in the five groups shown for the membrane blot for PKA and PKC. Supplementary pictures are not informative.
- Instead of a table, the bacterial profiles could be presented as pie charts or bar graphs to see the differences among samples better.
- L310: What is the mechanism behind the action of KAE to stabilize the mitochondrial apparatus in spermatozoa?
- -L329: is KAE preventing the production of ROS or simply scavenging to avoid the build-up of ROS amounts in spermatozoa?
- L340-41: HOW KAE treatment will increase SOD activity in spermatozoa?
- L348: How will KAE decrease H2O2 or hydroxyl radical levels in spermatozoa?
- L378-380: The action of KAE on membrane-bound ATPases was not demonstrated in spermatozoa. Thus, it is essential to conduct new experiments to determine whether KAE's effect in cryopreserved spermatozoa partially occurs by modulating these ATPases.
Author Response
Reviewer 2
1. No functional studies compared the fertilizing ability to demonstrate whether the KAE treatment is a good addition to the cryopreservation semen extender. For instance, incubation of frozen-thawed spermatozoa at 37C for two hours (Thermoresistance test) to determine sperm viability and evaluation of sperm capacitation and acrosome reaction should be presented.
Thank you for this recommendation. The experiments were repeated with frozen samples, that were subsequently subjected to the thermoresistance test as requested. Data are now presented pre-as well as post-thermoresistance test.
2. The fact that there is less protein in immunoblots from frozen-thawed spermatozoa compared to fresh or treated ones doesn't mean that the antioxidant treatment improves the expression of proteins. Instead, there is a prevention of losing these proteins because of membrane damage during frozen-thawing. It is more likely that KAE prevents deterioration of the plasma membrane (e.g. decreasing lipid peroxidation) and, thus, avoids the leaking of proteins from the spermatozoon.
Thank you for this interesting question. Prior to PAGE, each sample is subjected to protein quantification and subsequently normalized so that identical protein amount is loaded into the gel. Post-PAGE protein uniformity is validated by taking a picture of the gel before the transfer and subsequently checked again on the membrane using Ponceau-S. The presence of the proteins is finally validated by using Beta-Actin as the housekeeping protein.
We may agree that being a lipophilic substance, KAE will affect the membranous structures, and hence prevents the loss of proteins. We have avoided to relate to the changes as “improvement” and used rather “stabilization”.
3. The rationale to include the evaluation of bacterial profiles should be stated. This study seems to merit another complete research rather than being a small part of this manuscript.
Thank you for this remark. We have added more information to strengthen the rationale for bacteriological analysis. In this paper, we wanted to get a more complex view on the properties of KAE. While we appreciate the thought of a separate manuscript purely based on bacteriology, we believe the data would not have a strong impact by being presented in a single-topic paper.
4. The discussion is a mere repetition of results, and no molecular mechanisms are described to support the role of KAE as a protector of spermatozoa during cryopreservation.
Thank you for this recommendation. The section has been revised and me strived to add more comprehensive hypotheses towards the proposed mechanisms of action.
5. L80: Ref 16 does not talk about bovine cryopreservation. Please change it for a specific review on bovine semen (PMID: 37001221). Also, pioneer work on antioxidants and bovine cryopreservation and the effects of antioxidants on bull sperm function (motility and capacitation) by Dr. Beconi's group has been omitted in the introduction and discussion.
Thank you for this suggestion. Three studies by Dr. Beconi on natural antioxidants as well as the role of ROS on the sperm activation have been added and discussed.
6. It is easier for the reader to see the data as bar graphs instead of tables. So many numbers and variables make the analysis very difficult. For instance, It would be more illustrative to clearly present the motility, membrane and acrosome integrity and mitochondrial activity as bar graphs to show differences among samples in Table 1. For these parameters, particularly for sperm motility and acrosome integrity, the differences between the frozen-thawed spermatozoa control or treated with KAE are not clinically relevant (differences less than 30%).
Thank you for this suggestion. All data are now presented as bar graphs.
7. It is unclear why the effect of 50 uM KAE to protect DNA integrity is lower than that of the other concentrations. How do you explain this decrease in protection?
The phenomenon may be related to KAE “auto-oxidation”, which has been recently observed in somatic cell lines. High KAE concentrations will lead to the biomolecule to exhibit a more pro-oxidant behavior with a special affinity to DNA. This has been further discussed in the paper.
8. The KAE treatment does not lower premature sperm capacitation and ROS levels in spermatozoa.
According to the collected data, lower levels of capacitated spermatozoa as well as levels of global ROS/specific ROS classes have been observed following KAE treatment.
9. Western blots are not well presented, as it is difficult to see the molecular mass of the bands showing. The blots are cut; thus, it is not possible to see if other bands are also present (like for CatSper 2 or beta-actin). These two proteins are not found as doublets; therefore, these blots suggest a problem with antibody specificity and immunoblotting procedure. Was the same band stripped to blot for all proteins? The background is not even in the five groups shown for the membrane blot for PKA and PKC. Supplementary pictures are not informative.
Thank you for this comment. Before we did our experiments, a test run of each antibody was performed. We were lucky in selecting antibodies that by and large produced one band, which made us confident in cutting the membranes prior to the blocking procedure. One membrane was always cut in two pieces carrying a lower molecular weight and a higher molecular weight protein. The membrane carrying CatSper2 was stripped and re-used for PKA, and subsequently for Beta-actin, the other piece was used for CatSper1, NBC and PKC. The membranes were stripped with the Re-Blot Plus solution. Each protein did give us a different background, usually with a faint phantom banding, depending on the power of the antibodies/incubation time with ECL. The ChemiDoc visualizing equipment allows us to modify the intensity of the final picture live, since we usually take pictures on the Auto-mode and adjust the final photo depending on the intensity of the bands.
10. Instead of a table, the bacterial profiles could be presented as pie charts or bar graphs to see the differences among samples better.
Thank you for this recommendation. The data are now displayed as bar charts.
11. L310: What is the mechanism behind the action of KAE to stabilize the mitochondrial apparatus in spermatozoa?
We hypothesize that KAE may exhibit different mechanisms in relation to the mitochondria, namely:
Stabilization of the mitochondrial membrane similarly to the cytoplasmic membrane
Quenching of ROS overproduced by the mitochondria as a cause of cryogenic injury
Maintenance of the BAX/Bcl-2 ratio
12. L329: is KAE preventing the production of ROS or simply scavenging to avoid the build-up of ROS amounts in spermatozoa?
Thank you for this interesting question. We believe that both phenomena may be true. Obviously, the primary one will be neutralization of ROS that are already produced. However, if KAE is incorporated into the membranes, stabilizing them, then the chances of oxidative lipoperoxides and other reactive by-products of lipid peroxidation being formed are lower. Also, if KAE quenches excessive superoxide and hydrogen peroxide, then the risk of them interacting and hence creating the hydroxyl radical are lower as well.
13. L340-41: HOW KAE treatment will increase SOD activity in spermatozoa?
Thank you for this remark. Perhaps, the working is not correct. KAE may most likely stabilize the enzyme and provide protection to its oxidative inactivation. Nevertheless, if such result is compared to a positive control, compromised by whatever challenging situation, then the enzymatic activity may seem incteased.
14. L348: How will KAE decrease H2O2 or hydroxyl radical levels in spermatozoa?
Thank you for this interesting question.
In case of hydrogen peroxide, if KAE does quench excessive superoxide, then SOD will not dismutate into hydrogen peroxide. Furthermore, hydrogen peroxide may be quenched by KAE directly.
In case of hydroxyl radical, if the levels of superoxide and hydrogen peroxide are not relevantly high enough to interact with copper or iron withing the Fenton/Haber-Weiss reaction, hydroxyl radical will not be generated in relevant levels. Furthermore, hydroxyl radical may be quenched by KAE directly.
15. L378-380: The action of KAE on membrane-bound ATPases was not demonstrated in spermatozoa. Thus, it is essential to conduct new experiments to determine whether KAE's effect in cryopreserved spermatozoa partially occurs by modulating these ATPases.
Thank you for this very interesting question. We followed the recommendation and assessed the activities of Mg2+-ATPase, Ca2+-ATP-ase and Na+/K+-ATPase following the procedure by Zhao and Buhr, following a modified version of the original protocol by Breitbart. It was revealed that while Ca2+-ATPase remained relatively unaffected by KAE, Mg2+-ATPase was seen to respond, while the prime enzyme to be affected by KAE was aNa+/K+-ATPase
Round 2
Reviewer 1 Report
Comments and Suggestions for Authors
Thanks for author’s efforts to improve quality of the article.
Author Response
Reviewer 1
Thanks for author’s efforts to improve quality of the article.
Thank you so much for your positive feedback.
Reviewer 2 Report
Comments and Suggestions for Authors
I thank the authors for making efforts to improve the manuscript. However, there are still important flaws that make the study of questionable scientific value. Important problems are the inappropriate statistical tools used, lack of essential experiments (e.g. lipid peroxidation determination), and lack of clinical relevance based on the minimal differences observed in almost all parameters studied by the addition of KAE. Unnecessary results (bacterial profile) that do not provide meaningful novel information as antibiotics are a standard practice in the semen cryopreservation industry.
Specific comments:
L26: what is a ‘capacitation motif’? Please, explain.
L43: what do you mean by ‘stabilization of the sperm metabolism during cryopreservation?
L77-80: Reference 10 is a review paper and mentions the role of PKC in bull acrosome reaction. A better reference was published in 2005 by O’Flaherty et al (PMID: 16517508), which was the first time that PKA and PKC were associated with bull sperm capacitation.
L136-156 and Fig. 1: The statistical analysis of the thermoresitance test results is inappropriate. The authors used a one-way ANOVA followed by a Dunnett’s test. In this test, there are two variables time and treatment, thus two-way ANOVA followed by post-hoc test should be used. In any case, the thermoresistance test results, although statistically significant using an inappropriate test in some cases, did not reveal an important increase in sperm quality in terms of motility, membrane integrity, and intact acrosomes by the addition of KAE. The difference between samples cryopreserved with or without KAE is approximately less than 19%. With this result, it is arguable that this small increase will benefit the bovine semen cryopreservation industry.
Results in Figs. 2 and 3 are also wrongly analyzed statistically.
Western blots: The authors claimed that the antibodies used were specific for each protein under study; however, some of them show several bands. The quality of the representative blots is very poor for some of the proteins. The method used to calculate the relative intensities of the bands should be explained. Moreover, as written in L262, it is incorrect to refer to CatSper1 (or other proteins mentioned in this manuscript) expression in spermatozoa, since these cells lost the ability to translate genes into proteins. The correct term will be protein amount.
Bacterial profiles: Visually, the results have been improved from the previous version. However, it is still very difficult to compare the different experimental groups. Clearly, the use of antibiotics in the semen extender explains the low bacterial count level in cryopreserved compared to fresh samples. However, the effect of KAE is not visible as the differences are very small according to the authors' analysis, using that scale is Fig 9A.
Knowing that the bovine semen extender contains antibiotics to reduce bacterial contamination, this reviewer still thinks that there is no rationale to include the bacterial profile in this study.
Discussion
L360-366: the assumption that KAE interacts with the sperm plasma membrane is not supported by data since lipid peroxidation was not determined in this study.
L372-373: reference 46 is wrongly cited here as it does not deal with ATP synthesis, calcium homeostasis, or ROS overproduction.
Author Response
Reviewer 2
L26: what is a ‘capacitation motif’? Please, explain.
Thank you for this question. It is essentially a capacitation pattern; we just used a synonym for the word. We switched it back to “pattern” for more clarity.
L43: what do you mean by ‘stabilization of the sperm metabolism during cryopreservation?
Thank you for this question. By that statement we mean that through the protection of the mitochondrial membrane potential against cryodamage, the metabolism is more stabilized in comparison to the cryopreserved control as shown by the activity of ATPases coupled with a lower ROS leakage. We rephrased the sentence for more clarity.
L77-80: Reference 10 is a review paper and mentions the role of PKC in bull acrosome reaction. A better reference was published in 2005 by O’Flaherty et al (PMID: 16517508), which was the first time that PKA and PKC were associated with bull sperm capacitation.
The references were changed. Thank you.
L136-156 and Fig. 1: The statistical analysis of the thermoresitance test results is inappropriate. The authors used a one-way ANOVA followed by a Dunnett’s test. In this test, there are two variables time and treatment, thus two-way ANOVA followed by post-hoc test should be used. In any case, the thermoresistance test results, although statistically significant using an inappropriate test in some cases, did not reveal an important increase in sperm quality in terms of motility, membrane integrity, and intact acrosomes by the addition of KAE. The difference between samples cryopreserved with or without KAE is approximately less than 19%. With this result, it is arguable that this small increase will benefit the bovine semen cryopreservation industry.
Thank you for this important suggestion. In accordance with the suggestion, for the sperm quality parameters assessed prior to and following the thermoresistance test, Two-Way ANOVA and Tukey’s post hoc test were selected. The remaining sperm characteristics were processed using One-Way ANOVA and Tukey’s post hoc test to unify the test for comparisons. New graphs with updated statistical differences have been added to the text.
With respect to the second question, ejaculates for this study were obtained from highly fertile Holstein stud bulls, bred for the purposes of insemination doses. Such semen samples traditionally present with a supreme quality and a higher tolerance to cryodamage. These molecular properties, coupled with a technologically well-managed and automated cryopreservation procedure used in this study may lie behind a relatively small improvement of several sperm quality characteristics in the presence of KAE in comparison with the cryopreserved control. Bull semen quality depends on a wide array of factors, including age, season, or breed. At the same time, different cryopreservation protocols are available for the breeder, from a traditional slow and/or manual freezing procedure to a computer-assisted programmed cryopreservation or vitrification. As such, we may speculate that more prominent protective effects of KAE might have been observed in the case of ejaculates obtained from older breeding bulls or beef breeds diluted and cryopreserved using different freezing techniques. This aspect has been added to the Discussion section as a limitation of the study.
Results in Figs. 2 and 3 are also wrongly analyzed statistically.
Updated figures stemming from two-way ANOVA have been added to the text. Thank you.
Western blots: The authors claimed that the antibodies used were specific for each protein under study; however, some of them show several bands. The quality of the representative blots is very poor for some of the proteins. The method used to calculate the relative intensities of the bands should be explained. Moreover, as written in L262, it is incorrect to refer to CatSper1 (or other proteins mentioned in this manuscript) expression in spermatozoa, since these cells lost the ability to translate genes into proteins. The correct term will be protein amount.
Thank you very much for your questions. We have included photos from previous standardization western blots for the selected antibodies to confirm their specificity. We stopped using streptactin for the visualization of the bands pretty early on since even its very low concentrations were overpowering the actual protein bands (as seen in the NBC blot). The faint banding may be a consequence of the ability of the ChemiDoc to actually allow us to increase or decrease the exposition of the membranes in order to acquire the best deepness of the target bands. We have also added the principle of the relative quantification of proteins used for the calculations. Finally, we exchanged the term “protein expression” into “protein levels” or “protein amounts”.
Bacterial profiles: Visually, the results have been improved from the previous version. However, it is still very difficult to compare the different experimental groups. Clearly, the use of antibiotics in the semen extender explains the low bacterial count level in cryopreserved compared to fresh samples. However, the effect of KAE is not visible as the differences are very small according to the authors' analysis, using that scale is Fig 9A.
Knowing that the bovine semen extender contains antibiotics to reduce bacterial contamination, this reviewer still thinks that there is no rationale to include the bacterial profile in this study.
Thank you for this opinion. The whole bacteriological section has been deleted.
L360-366: the assumption that KAE interacts with the sperm plasma membrane is not supported by data since lipid peroxidation was not determined in this study.
Thank you for this intriguing insight. Lipid peroxidation has been already assessed in our previous study on KAE based on the same freezing procedure (https://doi.org/10.3390/stresses3040047). Since we already know that KAE administration led to a lower extent of lipid peroxidation, alongside a stabilization of the protein levels of membrane-bound HSP90 and HSP70, we may speculate that KAE indeed interacts with the sperm membrane, and agree on this with your hypothesis from your previous review on this paper (“The fact that there is less protein in immunoblots from frozen-thawed spermatozoa compared to fresh or treated ones doesn't mean that the antioxidant treatment improves the expression of proteins. Instead, there is a prevention of losing these proteins because of membrane damage during frozen thawing. It is more likely that KAE prevents deterioration of the plasma membrane (e.g. decreasing lipid peroxidation) and, thus, avoids the leaking of proteins from the spermatozoon”).
L372-373: reference 46 is wrongly cited here as it does not deal with ATP synthesis, calcium homeostasis, or ROS overproduction.
Thank you for this comment. The reference has been deleted.
Round 3
Reviewer 2 Report
Comments and Suggestions for Authors
I thank the authors for taking into consideration the comments and improving the manuscript.